# Overexpression of *ZmSTOP1-A* Enhances Aluminum Tolerance in *Arabidopsis* by Stimulating Organic Acid Secretion and Reactive Oxygen Species Scavenging

**DOI:** 10.3390/ijms242115669

**Published:** 2023-10-27

**Authors:** Chan Liu, Xiaoqi Hu, Lei Zang, Xiaofeng Liu, Yuhui Wei, Xue Wang, Xinwu Jin, Chengfeng Du, Yan Yu, Wenzhu He, Suzhi Zhang

**Affiliations:** 1Key Laboratory of Biology and Genetic Improvement of Maize in Southwest China of Agricultural Department, Ministry of Agriculture, Maize Research Institute, Sichuan Agricultural University, Chengdu 611130, China; liuchan.siu@foxmail.com (C.L.); huhuhxq@163.com (X.H.); zangleisiau@163.com (L.Z.); lxf11062023@163.com (X.L.); weiyuhui371102@163.com (Y.W.); 18583292575@163.com (X.W.); 13219027631@163.com (X.J.); duyeye1936551448@163.com (C.D.); yuyanbigfish@163.com (Y.Y.); 2Crop Research Institute, Sichuan Academy of Agricultural Sciences, Chengdu 610066, China; wenzu-he@163.com

**Keywords:** *Zea mays*, aluminum toxicity, *STOP1*, low pH, ZmSTOP1-A

## Abstract

Aluminum (Al) toxicity and low pH are major factors limiting plant growth in acidic soils. Sensitive to Proton Rhizotoxicity 1 (STOP1) transcription factors respond to these stresses by regulating the expression of multiple Al- or low pH-responsive genes. *ZmSTOP1-A*, a STOP1-like protein from maize (*Zea mays*), was localized to the nucleus and showed transactivation activity. *ZmSTOP1-A* was expressed moderately in both roots and shoots of maize seedlings, but was not induced by Al stress or low pH. Overexpression of *ZmSTOP1-A* in *Arabidopsis Atstop1* mutant partially restored Al tolerance and improved low pH tolerance with respect to root growth. Regarding Al tolerance, *ZmSTOP1-A/Atstop1* plants showed clear upregulation of organic acid transporter genes, leading to increased organic acid secretion and reduced Al accumulation in roots. In addition, the antioxidant enzyme activity in roots and shoots of *ZmSTOP1-A/Atstop1* plants was significantly enhanced, ultimately alleviating Al toxicity via scavenging reactive oxygen species. Similarly, ZmSTOP1-A could directly activate *ZmMATE1* expression in maize, positively correlated with the number of Al-responsive GGNVS *cis*-elements in the *ZmMATE1* promoter. Our results reveal that ZmSTOP1-A is an important transcription factor conferring Al tolerance by enhancing organic acid secretion and reactive oxygen species scavenging in *Arabidopsis*.

## 1. Introduction

Acid soil accounts for 50% of potential arable land worldwide, yet can severely inhibit plant growth and reduce crop yields [1]. The primary limiting factor in acid soil is aluminum (Al) toxicity [2]. Below pH 5.0, Al solubilizes into the phytotoxic Al^3+^ form, which damages plant root structure, impairs water and nutrient uptake, and ultimately inhibits normal plant growth [3,4]. Elucidating the genes involved in Al tolerance mechanisms is therefore crucial for improving crop yields through genetic approaches.

Generally, external exclusion and internal tolerance are two main mechanisms for plants to cope with Al stress [5]. The external exclusion refers to the secretion of organic acid from the rhizosphere, of which chelate Al^3+^ prevents its entry into the root. MATE (multidrug and toxic compound extrusion) and AlMT1 (aluminum-activated malate transporter 1), two kinds of organic acid transporter of the external exclusion mechanism, are crucial for Al tolerance in many crops [6,7]. Internal tolerance involves sequestration of excessive Al into the vacuoles or immobilization in the cell wall. Additionally, recent evidence indicates that scavenging of reactive oxygen species (ROS) also plays an important role in Al detoxification [8]. Some genes, including *ZmAT6* (Aluminum transporter 6), *AtGST* (Glutathione S-transferase) and *AtPrx64* (Peroxidases 64), have been implicated in removing ROS during Al stress [9,10,11]. To date, many factors including transcription factors, organic acid transporter, ABC transporter, Mg transporter, enzymes related to cell wall components or ROS scavenging, phytohormones, and small peptide have been reported to play roles in the Al stress response across plant species [12]. Transcription factors, especially the STOP1-like class, are key regulators which charge the expression of many Al-responsive genes and play a central role in Al detoxification [13].

AtSTOP1, a C2H2 zinc finger transcription factor first identified in *Arabidopsis* [14], has since been found in several crops, including rice *(Oryza sativa*) [15], rice bean (*Vigna umbellata*) [16], and barley (*Hordeum vulgare*) [17]. AtSTOP1 is a key regulator preventing Al toxicity by controlling the expression of 24 Al-responsive genes, including *AtALMT1* and *AtALS3* (Aluminum sensitive 3) [13]. Its rice homolog OsART1 (Al resistance transcription factor 1) similarly targets 31 genes, including *OsFRDL4* and *OsSTAR2* (Sensitive to aluminum rhizotoxicity 2) for Al detoxification. Notably, most of these AtSTOP1/OsART1-regulated genes contain the *cis*-element GGN(T/g/a/C)V(C/A/g)S(C/G) in their promoters [15,18]. Beyond Al stress, the STOP1-like transcription factor also responds to low pH by regulating a distinct set of genes, such as *AtSTOP2*, *AtGDH1* (Glutamate dehydrogenase 1), *AtCIPK23* (CBL-Interacting protein kinase 23), and *AtNRT1.1* (Nitrate transporter 1.1) [13,19,20]. However, some STOP1 homologs are insensitive to low pH [15,21,22].

Maize is one of the most important food crops, yet its yield drops dramatically when grown on acidic soils [23]. The genetic mechanisms underlying Al toxicity response of maize remain poorly understood. To date, only six genes have been reported to contribute to Al tolerance in maize, by facilitating citric acid efflux (*ZmMATE1* and *ZmMATE6*) [24,25], ROS elimination (*ZmALDH1* and *ZmAT6*) [11,26], cell wall fixation of Al (*ZmXTH1*) [27], and auxin transport (*ZmPGP1*) [28]. In this study, *ZmSTOP1-A* was first identified in maize, and its Al-responsive expression pattern was explored. Most importantly, the roles of *ZmSTOP1-A* in Al- and low pH tolerance were genetically investigated through ectopic expression in the *Arabidopsis* mutant *Atstop1*. The mechanisms underlying *ZmSTOP1-A*-mediated Al tolerance were also examined by surveying the expression of Al-responsive genes and measuring key physical indexes and Al content in the tested plants. This study provides important insights into how the maize *ZmSTOP1-A* responds to both Al toxicity and low pH. The findings expanded our understanding of the role STOP1-like transcription factors play in enabling crop species to adapt to acidic environments.

## 2. Results

### 2.1. Identification and Phylogenetic Analysis of ZmSTOP1-A

Based on sequence homology to the proteins of AtSTOP1 and OsART1, six homologs, ZmSTOP1-A, ZmSTOP1-B, ZmSTOP1-C, ZmSTOP1-D, ZmSTOP1-E, and ZmSTOP1-F, were identified in maize. As with other STOP1-like transcription factors, the ZmSTOP1 proteins, except for ZmSTOP1-D, contain four highly conserved putative C2H2 zinc finger domains characteristic of STOP1-like proteins (Appendix A). Phylogenetic analysis showed that ZmSTOP1-A and ZmSTOP1-B clustered on the same branch, closely related to SbSTOP1-d, three TaSTOP1s, and AtSTOP1. In a branch above ZmSTOP1-A, ZmSTOP1-C and ZmSTOP1-D were most closely related to OsART1 and SbSTOP1-a, respectively. Meanwhile, ZmSTOP1-E and ZmSTOP1-F were in the same branch as AtSTOP2 and OsART2, adjacent to SbSTOP1-b and SbSTOP1-c (Figure 1). This reveals that STOP1-like proteins have complex evolutionary relationships characterized by both conservation and divergence.

### 2.2. Expression Pattern of ZmSTOP1-A

To examine whether these six *ZmSTOP1-like* genes were responsive to Al stress, their expression was analyzed by quantitative real-time PCR in maize seedling roots. As shown in Appendix A, *ZmSTOP1-like* gene expression was constitutive and not induced by Al toxicity. However, ZmSTOP1-A showed relatively high expression and was chosen for further Al-response analyses. 

*ZmSTOP1-A* exhibited similar expression patterns in both roots and shoots with or without Al stress (Figure 2A). Further testing found that Al exposure did not affect *ZmSTOP1-A* expression in different root segments (0–5 mm, 5–10 mm, 10–20 mm) (Figure 2B), nor did low pH treatment (Figure 2C). *Arabidopsis* plants transformed with a *ZmSTOP1-A* promoter-GUS construct confirmed this constitutive expression pattern in roots, leaves, and stems. These results demonstrate that *ZmSTOP1-A* is constitutively expressed in both roots and shoots, and not induced by Al toxicity and low pH (Figure 2D,E).

### 2.3. Subcellular Localization and Transcriptional Activation of ZmSTOP1-A

When transiently expressed in onion epidermal cells, ZmSTOP1-A::GFP fluorescence was only observed in the nucleus (Figure 3A) in comparison with 35S::GFP control, which localized to both the nucleus and plasma membrane. This indicates that ZmSTOP1-A is a nucleus protein.

The transcriptional activation ability of ZmSTOP1-A was also verified in the yeast system. As shown in Figure 3B, yeast cells transformed with empty vector p*GBKT7* did not grow on screening medium. However, yeast cells expressing p*GBKT7-ZmSTOP1-A* grew normally and turned blue after X-gal addition, indicating that ZmSTOP1-A has transcriptional activation capability, a defining property of transcription factors.

### 2.4. Ectopic Expression of ZmSTOP1-A Enhanced Al and Low-pH Tolerance in Transgenic Arabidopsis

Whether *ZmSTOP1-A* is also involved in Al toxicity and low pH stress in maize is unclear as yet. To examine this possibility, *ZmSTOP1-A* was transformed to *Atstop1* mutant, and highly expressed transgenic lines *ZmSTOP1-A/Atstop1* #5 and #8 (Appendix A) were selected for genetic complementary evaluation. Under Al stress (pH 4.9 + Al), the relative root growth (RRG) of *ZmSTOP1-A/Atstop1* #5 and #8 was slightly lower than wild type (WT) but significantly higher than *Atstop1*, indicating that *ZmSTOP1-A* could partially restore the Al-induced root growth inhibition in *Atstop1* (Figure 4A,B). Similarly, at pH 4.9, *ZmSTOP1-A/Atstop1* (#5 and #8) fully restored RRG of *Atstop1* to WT levels (Figure 4A,C). These results demonstrate that *ZmSTOP1-A* facilitates *Arabidopsis* plants to antagonize both Al toxicity and low pH.

It has been shown that *Arabidopsis* AtSTOP1 responds to Al toxicity and low pH by regulating the expression of different set of genes [13]. Thus, the expression of *AtALMT1*, *AtMATE,* and *AtALS3* (Al-responsive), and *AtSTOP2*, *AtGDH1*, *AtGDH2*, *AtNRT1.1,* and *AtCIPK23* (low-pH-responsive) were examined here. Under Al stress, Al-responsive genes expression was partially restored in all *ZmSTOP1-A/Atstop1* plants compared to *Atstop1* (Figure 5). Under low-pH conditions, *AtGDH1*, *AtGDH2,* and *AtNRT1.1,* but not *AtSTOP2* and *AtCIPK23,* expression fully recovered in *ZmSTOP1-A/Atstop1* plants (Figure 5). These expression patterns align with the RRG of *ZmSTOP1-A/Atstop1* plants under Al toxicity and low-pH conditions.

### 2.5. ZmSTOP1-A Partially Rescued Al Tolerance of Atstop1 Attributed to the Secretion of Organic Acids

The primary mechanism for crops to detoxify Al is extrusion of organic acids from the root apex. As expected, under Al stress, the secretion of malate and citrate was significantly higher in *ZmSTOP1-A/Atstop1* plants compared to *Atstop1*, but still lower than WT (Figure 6A,B). This aligns with the expression of *AtALMT1* and *AtMATE*. Furthermore, Al content in *ZmSTOP1-A/Atstop1* roots was higher than WT but lower than *Atstop1* (Figure 6C). Hematoxylin staining confirmed these findings (Figure 6D). Together, these results indicate that organic acid extrusion is a key strategy by which ZmSTOP1-A alleviates Al toxicity of *Atstop1*. 

### 2.6. ZmSTOP1-A Responds to Al-Induced Oxidative Stress in the Atstop1 Mutant

Al toxicity can induce ROS production, and scavenging of these ROS mitigates Al stress. Thus, the activities of antioxidant enzymes, SOD, CAT, POD, and APX, which eliminate ROS, were examined. Under Al stress, O_2_^−^ accumulation was higher in all plant roots, but O_2_^−^ content in *ZmSTOP1-A/Atstop1* was similar to WT and lower than in *Atstop1* (Figure 7A). In contrast, H_2_O_2_ increased in roots upon Al treatment, with no difference between +Al or −Al conditions (Figure 7B). Meanwhile, SOD, CAT, POD, and APX activities increased in Al-stressed roots. In *ZmSTOP1-A/Atstop1* plants, SOD, CAT, and POD activities fully recovered to WT levels, except for APX, which was similar across the tested plants (Figure 7C–F). The same pattern also occurred in shoots (Appendix A). This finding indicates that SOD, CAT, and POD preferentially contribute to scavenging Al-induced ROS, particularly O_2_^−^.

### 2.7. ZmSTOP1-A Directly Regulated ZmMATE1 Expression in Maize

Organic acid extrusion is the primary Al-tolerance mechanism in many plant species, with maize mainly secreting citrate via the important citrate transporter *ZmMATE1* [24]. In order to examine if ZmSTOP1-A directly regulated *ZmMATE1*, a transient promoter assay was performed in maize protoplasts. Co-transformation of promoter*^ZmMATE1^:LUC* and *35S:ZmSTOP1-A* enhanced LUC signal compared to promoter*^ZmMATE1^:LUC* alone (Figure 8B). This demonstrates that ZmSTOP1-A can activate *ZmMATE1* transcription. 

In addition, the *ZmMATE1* promoter was truncated into four (P1–P4) fragments for yeast one-hybrid analysis. P1 (−2020 to −1453), P2 (−1182 to −651), P3 (−504 to −285), and P4 (−285 to −1) contained 8, 10, 12, and 14 Al-responsive GGNVS *cis*-elements, respectively (Figure 8C). All yeasts grew normally at 100 ng/mL Aureobasidin A (AbA), but were inhibited at 400 ng/mL AbA. Compared to the pGADT7 control, yeasts with *ZmSTOP1-A* promoter fragment grew much better, declining in order of P4, P3, P2, and P1 (Figure 8D). This demonstrates that ZmSTOP1-A directly interacts with *ZmMATE1* promoter, with binding strength positively correlating with the GGNVS *cis*-element number.

## 3. Discussion

### 3.1. ZmSTOP1-A Is Involved in Both Al Stress and Low-pH Stress Responses

Al toxicity is the major limiting factor for plant growth in acid soils. Identification of Al-response genes and elucidation of their roles are essential to understand the molecular mechanisms of Al toxicity, and to develop molecular-assistant breeding approaches to improve crop yields under Al stress. The transcription factor AtSTOP1 and its rice homolog OsART1 are master regulators of over twenty Al-responsive genes in *Arabidopsis* and rice, respectively [14,15]. STOP1-like transcription factors have now been found across many plant species, where they are involved in both Al stress and low-pH stress responses, unless not yet investigated [19,21,29]. For example, AtSTOP1 was first identified in *Arabidopsis* from a proton-sensitive mutant, and later shown to mediate Al tolerance [14]. Similarly, GmSTOP1-1 and -3 in soybean [30], ScSTOP1 in rye [31], PpSTOP1 in moss, PnSTOP1 in poplar, LjSTOP1 in lotus, and NtSTOP1 in tobacco [19] have all been shown to function in both Al and low-pH stresses by complementing the *Arabidopsis Atstop1* mutant. In this study, *ZmSTOP1-A/Atstop1* plants also fully or partially restored the WT phenotype via regulating distinct Al- and low-pH-responsive genes (Figure 4 and Figure 5), consistent with findings on other STOP1-like transcription factors using this complementary system. 

Nevertheless, the downstream genes regulated by STOP1-like proteins differ somewhat between species in response to both Al and low-pH stresses. For example, *ZmSTOP1-A* rescues low-pH-sensitivity of *Atstop1* by upregulating *AtSTOP2*, *AtGDH1*, *AtGDH2*, *AtCIPK23,* and *AtNRT1.1* (Figure 5), while *NtSTOP1*, *PpSTOP1*, *EguSTOP1,* etc., act through *AtSTOP2*, *AtPGIP1*, and *AtCIPK23* [19,29]. *VuSTOP1* mediates via *AtSTOP2*, *AtGDH1*, *AtPGIP1*, and *AtCIPK23* [16]. Beyond that, STOP1-like gene expression patterns vary. Some are Al-inducible, like *OsART2*, *VuSTOP1*, *TaSTOP1-A*, *SbSTOP1s*, and *GmSTOP1-1* [16,21,22,30,32], while others, including *TaSTOP1-B*/-*D*, *AtSTOP1*, *AtSTOP2*, *EguSTOP1*, *HvATF1* (Al-tolerance transcription factor), and *OsART1,* are constitutively expressed regardless of Al exposure [14,15,17,21,29,33]. Similar variations exist under low pH (Appendix A). However, in fact, post-transcriptional modifications, including SUMOylation, ubiquitination, and phosphorylation, can modulate the stabilization of constitutively expressed STOP1-like protein, as demonstrated by AtSTOP1. These findings reveal that STOP1-like proteins have developed divergent function for plants species to adapt to fluctuating environments in acid soil.

In this study, *ZmSTOP1-A* expression was not affected by Al toxicity and low pH (Figure 2), similar to STOP1-like genes such as *AtSTOP1* and *OsART1* [14,15]. Considering the diverse post-translational regulation of AtSTOP1, such modifications likely also modulate ZmSTOP1-A. Future work could examine potential ZmSTOP1-A modification under Al stress by fusing fluorescent reporters or epitope tags for genetic and biochemical detection.

### 3.2. ZmSTOP1-A Enhanced Al Tolerance in Arabidopsis Mainly through External Extrusion Mechanism

Ectopic expression in the *Arabidopsis Atstop1* mutant is a widely used system to assay Al tolerance genes. For instance, under Al stress condition, *VuSTOP1* and *AtSTOP2* partially rescued *Atstop1* Al sensitivity by restoring the expression of *AtALS3* and *AtMATE*, but not *AtALMT1* [16,33]. Similarly, *PpSTOP1* partially recovered *AtALMT1*, *AtMATE,* and *AtALS3* expression and roots growth [19]. Here, *ZmSTOP1-A* also partially restored *AtALMT1*, *AtMATE,* and *AtALS3* expression in *Atstop1* under Al stress (Figure 5), correlating with increased organic acid secretion and lower Al accumulation (Figure 6). This is consistent with findings for *NtSTOP1* [19], suggesting that the STOP1-organic acid transporter module is highly conserved for Al detoxification across plants.

It is notable that *STOP1-like* genes in most species are predominantly expressed in roots with very low shoot levels, such as *OsART1*/*2* in rice [15,22], *VuSTOP1* in rice beans [16], *SbSTOP1S* in sweet sorghum [32], and *GmSTOP1s* in soybean [30]. In contrast, *ZmSTOP1-A* was expressed almost equally in roots and shoots of maize seedlings, implying a potential role for shoots in maize Al detoxification. *HvATF1* in barley follows this expression pattern, but its Al response was not explored [17]. In *ZmSTOP1-A/Atstop1* plants here, both the internal tolerance gene *AtALS3* and external exclusion genes *AtALMT1* and *AtMATE* were upregulated (Figure 5). Meanwhile, Al content in *ZmSTOP1-A/Atstop1* roots and shoots sharply declined to WT level, in comparison with *Atstop1* (Figure 6 and Appendix A). This reveals that external exclusion dominates in *ZmSTOP1-A/Atstop1* plants. The low Al absorption and root-to-shoot transport was insufficient to impact shoot Al content. This result suggests that *ZmSTOP1-A* may confer Al tolerance in maize primarily by regulating organic acid transporter expression, as demonstrated by other STOP1 transcription factors. For example, *AtSTOP1* regulates *AtALMT1* and *AtMATE* for organic acid secretion [13,14]; *OsART1* mediates citrate secretion by controlling *OsFRDL4* and *OsFRDL2* expression [15], and *CcSTOP1* directly regulates *CcMATE1* expression [34].

Additionally, *ZmMATE1* and *ZmMATE6* were previously reported to confer maize Al tolerance through citrate release [24,25]. Here, the transient assay in maize protoplasts and yeast one-hybrid analysis verified that ZmSTOP1-A directly binds to the Al-responsive GGNVS *cis*-element to activate *ZmMATE1* and detoxify Al (Figure 8B). Interestingly, ZmSTOP1-A activation of *ZmMATE1* positively correlated with the GGNVS *cis*-element number in the *ZmMATE1* promoter fragment (Figure 8D). This is consistent with findings for other organic transporters like HLALMT1 and OsFDRL4, which are activated by STOP1 proteins and show higher expression in Al-tolerant cultivars with more GGNVS elements [35,36]. Therefore, an increase in GGNVS elements in promoters of organic acid transporter genes could be a strategy for breeding Al-tolerant crops.

### 3.3. ZmSTOP1-A Eliminated the Al-Induced Oxidative Stress in Atstop1

Al toxicity triggers oxidative stress in many plants. Jay et al. found that Al toxicity caused excessive ROS (H_2_O_2_ and O_2_^−^) accumulation in rice roots, activating antioxidant enzymes including SOD, APX, POX, GR, CAT, DHAR, and MDHAR. Al-resistant varieties displayed stronger antioxidant defenses than sensitive ones [37]. Additionally, overexpression of the *Arabidopsis* peroxidase gene *AtPrx64* improved tobacco Al tolerance by scavenging ROS and reducing root Al accumulation [10]. Recently, *ZmAT6* overexpression in maize enhanced Al tolerance by dramatically increasing SOD and POD antioxidant enzyme activities [11]. In this study, *ZmSTOP1-A* also increased SOD, POD, and CAT activities in *Atstop1* roots and shoots, coordinately with the eliminated ROS (Figure 7 and Appendix A). This demonstrates that ROS scavenging assists Al detoxification, whether directly or indirectly.

In summary, ZmSTOP1-A, a C2H2 zinc finger transcription factor involved in Al toxicity and low-pH responses, was constitutively and equally expressed in maize roots and shoots. ZmSTOP1-A conferred tolerance to both Al toxicity and low-pH stress. Regarding Al toxicity, *ZmSTOP1-A* increased organic acid secretion and reduced Al accumulation by upregulating *AtALMT1* and *AtMATE* expression. It also enhanced antioxidant content to detoxify Al. In addition, *ZmMATE1* directly activated the maize organic acid transporter ZmSTOP1-A in a manner dependent on the GGNVS element. Overall, we proposed a model to describe the functional regulation of ZmSTOP1-A under aluminum stress and low-pH conditions (Figure 9). These results demonstrate that ZmSTOP1-A plays an important role in responding to Al toxicity and low pH by regulating organic acid secretion and ROS scavenging. 

## 4. Materials and Methods

### 4.1. Plant Materials and Culture Conditions

The Al-tolerant maize inbred line 178, which was screened from a large maize population [38], was used in this study. Maize seedlings were cultivated in nutrient solution under growth conditions as previously described [24,39]. The nutrient solution contained the following macronutrients (in millimoles): Ca, 3.53; K, 2.35; Mg, 0.85; NH_4_, 1.3; NO_3_, 10.86; PO_4_, 0.04; and SO_4_, 0.59, and micronutrients (in micromoles): BO_3_, 25; Cl, 596; Cu, 0.63; Fe-HEDTA, 77; MoO_4_, 0.83; Mn, 9.1; Zn, 2.3; and Na, 1.74. After 24 h of pretreatment, two-leaf stage seedlings were exposed to the same solution with or without 222 μM Al [KAl(SO_4_)_2_] (effective Al^3+^ 39 μM) at different pH (4.0, 4.5, 5.0, 5.5, 6.0, and 6.5) for 6 h. All treatments were performed in three replicates for RNA extraction [14]. Wild-type (WT) Col-0, *Atstop1* mutant (At1g34370, SALK_114108), and transgenic *Arabidopsis* were grown under conditions as previously described by Kobayashi et al. [14].

### 4.2. Phylogenetic Tree Construction

The phylogenetic tree of STOP1-like proteins from maize (ZmSTOP1-A, Zm00001d042686; ZmSTOP1-B, Zm00001d012260; ZmSTOP1-C, Zm00001d023558; ZmSTOP1-D, Zm00001d034783; ZmSTOP1-E, Zm00001d016911; ZmSTOP1-F, Zm00001d031895) and other species was constructed using the neighbor-joining method in MEGA 5.1.

### 4.3. RNA Extraction and Quantitative Real-Time RT-PCR

Total RNA extraction and quantitative RT-PCR were performed as previously described [11]. All primers were designed using Primer 5.0 software (Appendix A). *ZmGAPDH* and *AtACT2* were used as reference genes. Each sample was assayed in at least three technical replicates. 

### 4.4. Subcellular Localization

The full-length ORF of *ZmSTOP1-A*, excluding the stop codon, was fused to *GFP* in pCAMBIA2300 to generate the *ZmSTOP1-A::GFP* fusion construct. This *ZmSTOP1-A::GFP* vector and a *35S::GFP* control vector were transiently expressed in onion epidermal cells, as described previously by Li et al. [40].

### 4.5. GUS Staining Assay

A 2.1 kb region upstream of the start codon of *ZmSTOP1-A* was amplified and cloned into vector pCAMBIA3301. GUS staining was performed as previously described [41]. 

### 4.6. Determination of Organic Acids Secretion and Al Content

After Al stress treatment, root exudate organic acids concentrations in *Arabidopsis* were measured, and hematoxylin staining was performed as previously described [25]. Root and shoot samples were collected separately and Al content was qualified by inductively coupled plasma mass spectrometry (NexlON 2000 ICP-MS, PerkinElmer, Singapore).

### 4.7. Transient Expression Assay in Maize Protoplasts

As described by Li et al. [40], the reporter vector was generated by cloning the *ZmMATE1* promoter into the pBI221 vector. The effector vector was created by inserting the coding sequences of *ZmSTOP1-A*, driven by an ubiquitin promoter, into the *Pst*Ⅰ and *Bam*HⅠ sites of pBI221. The ubi::*GUS* was used as the internal vector. After 12–14 h culturing in the dark, the co-transformed maize 178 protoplasts containing the reporter vector, effecter vector, and internal vector (at a 2:1:1 molar ratio) were observed for LUC and GUS signal. The experiments were conducted with three independent replicates. 

### 4.8. Transcriptional Activity Detection and Yeast One-Hybrid Assay

For the yeast transactivation assay, the bait vector pGBKT7 with *ZmSTOP1-A* fused to the GAL4 DNA-binding domain (BD) was used to transform the Y2H gold yeast strain, as previously described [32]. 

For the yeast one-hybrid assay [42], the promoter region of *ZmMATE1* was amplified and cloned into the pAbAi vector. *ZmSTOP1-A* was fused to the *GAL4* activation domain (AD) in the pGADT7 vector. Finally, this pair of constructs was then introduced into the Y1H gold yeast strain. 

### 4.9. Determination of ROS Content and Antioxidant Enzyme Activity

O^2−^ and H_2_O_2_ content were individually determined following the method of Erich et al. [43]. Activities of SOD (EC 1.15.1.1), CAT (EC 1.11.1.6), APX (EC 1.11.1.11), and POX (EC 1.11.1.7) were carried out as previously described [37].

## Figures and Tables

**Figure 1 ijms-24-15669-f001:**
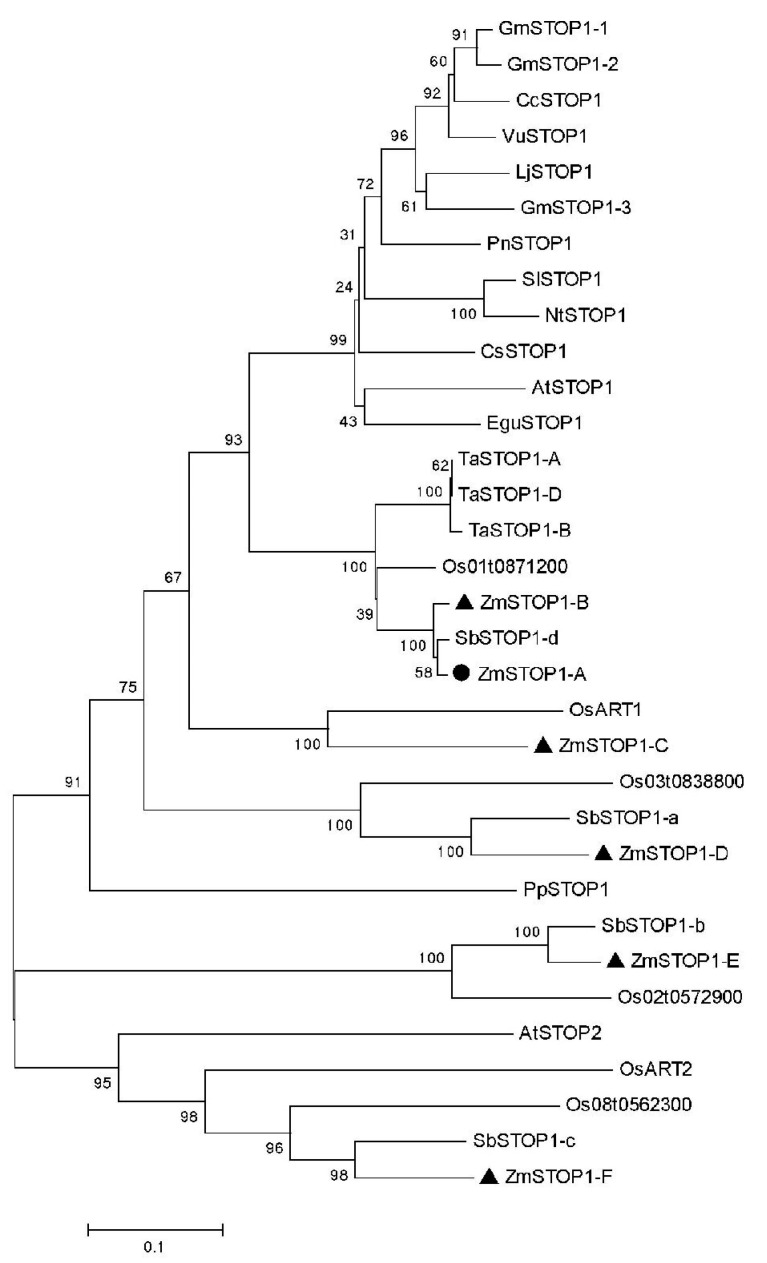
Phylogenetic tree of ZmSTOP1-like proteins. The phylogenetic tree was constructed based on an amino acid sequence alignment of STOP1 orthologs from several plant species.

**Figure 2 ijms-24-15669-f002:**
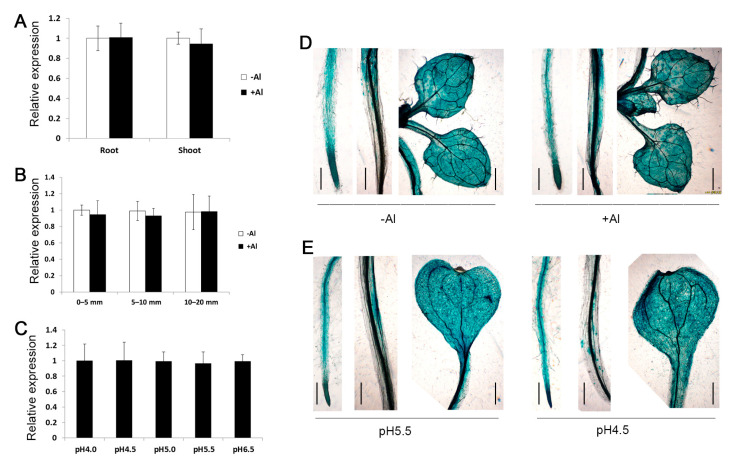
Spatial expression pattern of *ZmSTOP1-A* in maize seedlings. The expression of *ZmSTOP1-A* in different (**A**) organs, (**B**) root segments, and (**C**) under different pH. GUS staining of roots, stems, and leaves of *promoter^ZmSTOP1-A^::GUS* transgenic *Arabidopsis* plants under (**D**) Al stress and (**E**) low pH. Al stress was performed in the solution containing 0 or 222 μM Al [KAl(SO_4_)_2_] at pH 4.0 for 6 h. Mean values and SD (*n* = 3) are shown. At least two independent transgenic lines were used for GUS staining. Bars = 200 μm.

**Figure 3 ijms-24-15669-f003:**
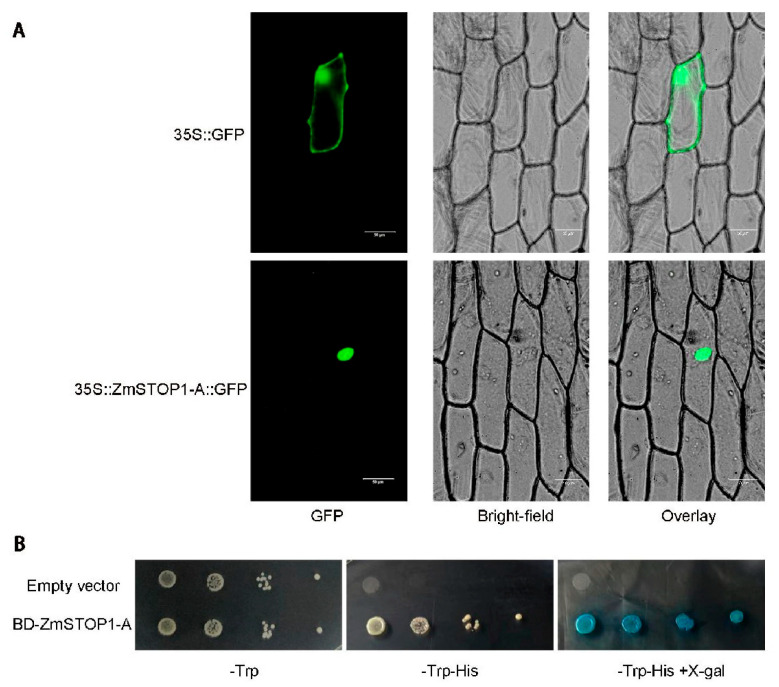
Subcellular localization and transactivation activity of ZmSTOP1-A. (**A**) *35S::GFP* (control) and *35S::ZmSTOP1-A::GFP* constructs were expressed in onion epidermal cells. Bars = 50 μm. (**B**) *β*-galactosidase activity was indicated by blue color using X-gal as the substrate.

**Figure 4 ijms-24-15669-f004:**
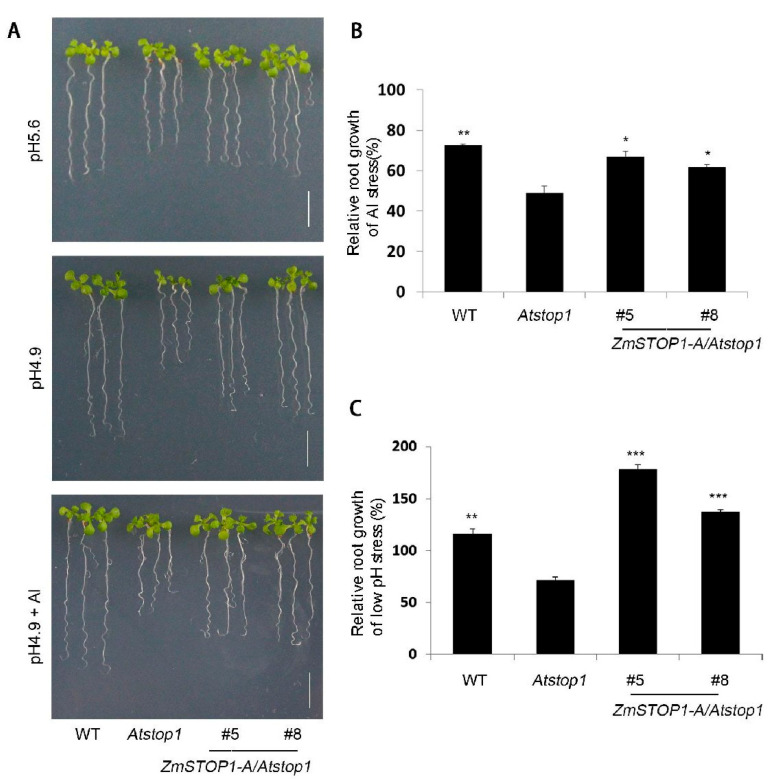
Phenotypes and relative root growth of *ZmSTOP1-A/Atstop1* under Al toxicity and low-pH stress. (**A**) Phenotypes of WT, *Atstop1,* and *ZmSTOP1-A/Atstop1* (#5 and #8) under normal condition (pH 5.6), Al stressed under low-pH condition (20 μM AlCl_3_, pH 4.9) and low-pH condition (pH 4.9) for 7 days. Seedlings grown at normal condition at pH 5.6 were used as control. Bars = 1 cm. (**B**) Relative root growth (RRG, %) of Al stress. (**C**) RRG (%) of low-pH stress. Means and SD (*n* = 10) are shown. Asterisks indicate significant differences compared to *Atstop1* (Tukey’s test; * *p* < 0.05; ** *p* < 0.01; *** *p* < 0.001).

**Figure 5 ijms-24-15669-f005:**
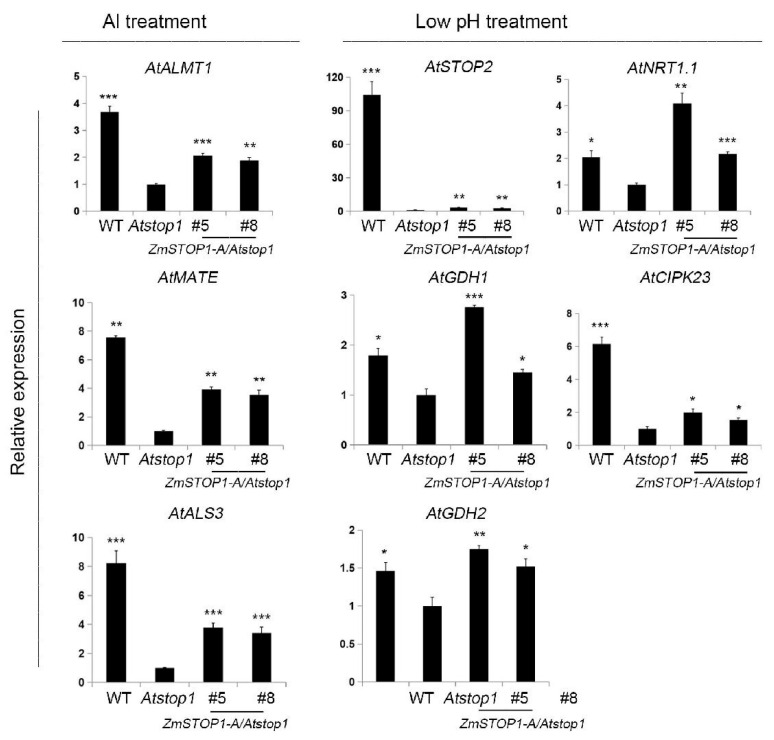
Expression of *ZmSTOP1-A* regulated genes under Al toxicity or low-pH stress. WT, *Atstop1*, and *ZmSTOP1-A/Atstop1* plants were exposed to 20 μM AlCl_3_ (pH 4.9) or pH 4.9 for 24 h. Al-responsive genes *AtALMT1*, *AtMATE*, *AtALS3,* and low-pH-responsive genes *AtSTOP2*, *AtGDH1*, *AtGDH2*, *AtNRT1.1*, and *AtCIPK23* were quantified by qRT-PCR using *AtACT2* as an internal control. Mean values and SD (*n* = 3) are shown. Asterisks indicate significant differences compared to *Atstop1* (Tukey’s test; * *p* < 0.05; ** *p* < 0.01; *** *p* < 0.001).

**Figure 6 ijms-24-15669-f006:**
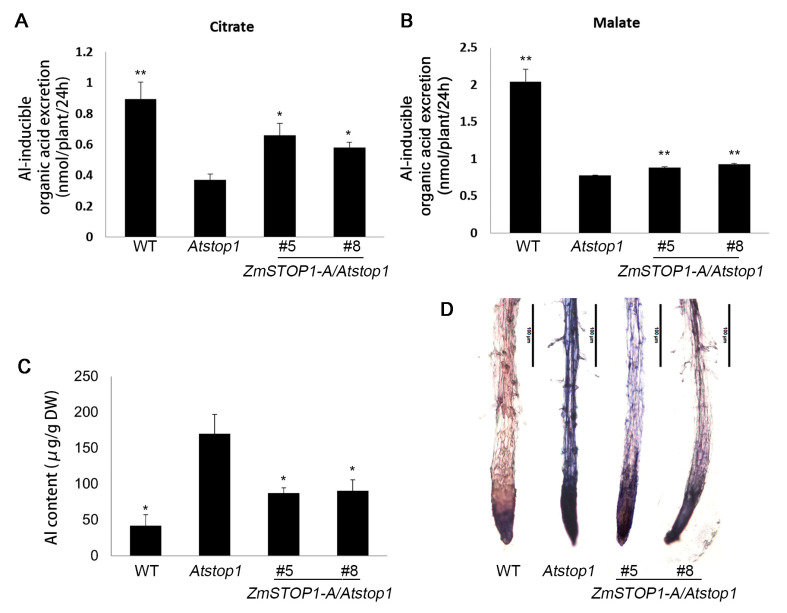
Overexpression of *ZmSTOP1-A* reduced Al accumulation in *Atstop1* mutant. Secretion of (**A**) citrate and (**B**) malate, (**C**) Al content, and (**D**) hematoxylin staining in roots of WT, *Atstop1* and *ZmSTOP1-A/Atstop1* plants after 24 h treatment with or without 20 μM AlCl_3_ (pH 4.9). Values in A, B, and C represent mean ± SD (*n* ≥ 10). Asterisks indicate significant differences compared to *Atstop1* (Tukey’s test; * *p* < 0.05; ** *p* < 0.01).

**Figure 7 ijms-24-15669-f007:**
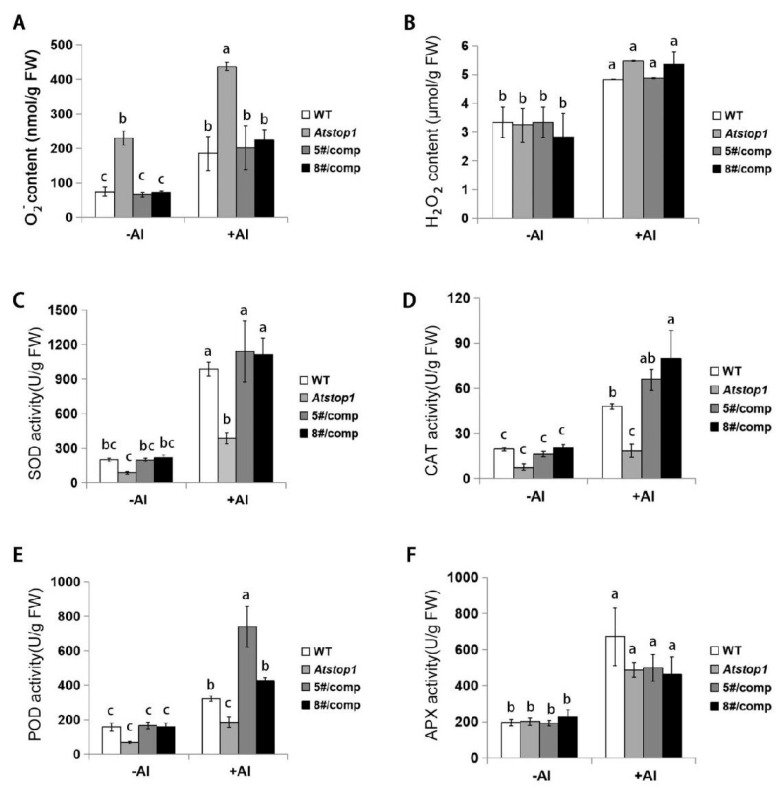
Effects of Al toxicity on reactive oxygen species in WT, *Atstop1*, and *ZmSTOP1-A/Atstop1* plants. Superoxide anion (O_2_^−^) (**A**) and hydrogen peroxide (H_2_O_2_) (**B**) content and activity of SOD (**C**), CAT (**D**), POD (**E**), and APX (**F**) in roots after 24 h exposure with 0 or 20 μM AlCl_3_ (pH 4.9). Values are mean ± SD (*n* ≥ 10) of three separated experiments. Different letters indicate significant difference (Tukey’s test, *p* < 0.05).

**Figure 8 ijms-24-15669-f008:**
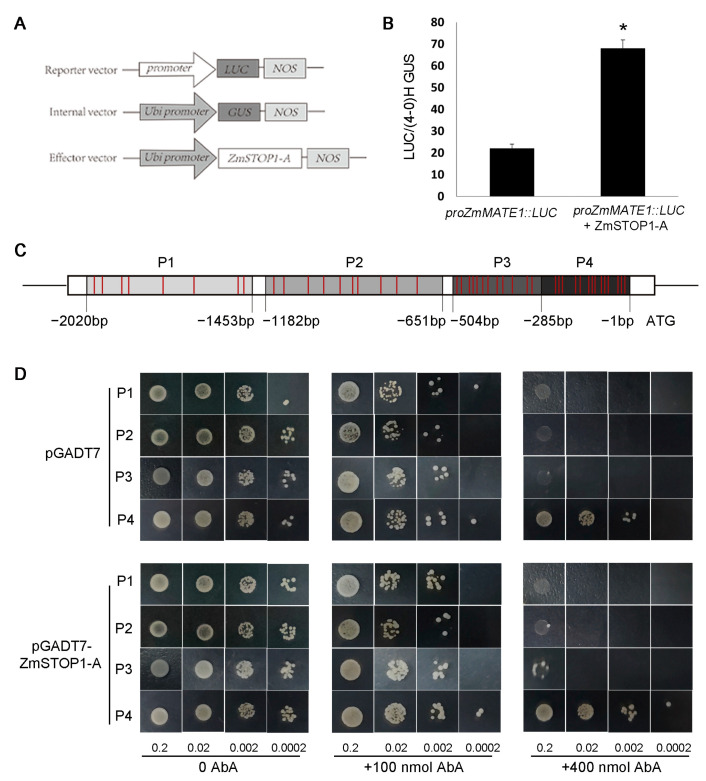
*ZmSTOP1-A* activated *ZmMATE1* expression by binding to its promoter fragments. (**A**) Schematic of transient expression vectors. (**B**) ZmSTOP1-A activation of *ZmMATE1* in maize protoplast assay. Values are means ± SD (*n* = 3). Asterisks indicate significant differences between with and without effector vectors (Tukey’s test, * *p* < 0.05). (**C**) Schematic diagram of *ZmMATE1* promoter bait fragments (P1–P4) used to construct reporter vectors. Red indicates GGNVS *cis*-elements. (**D**) Binding of ZmSTOP1-A to different *ZmMATE1* promoter fragments in yeast one-hybrid assay.

**Figure 9 ijms-24-15669-f009:**
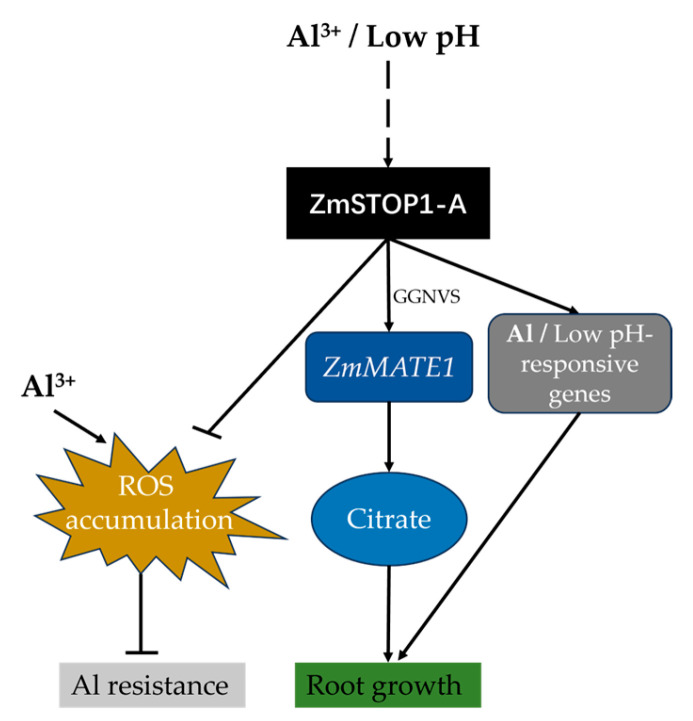
A hypothetical model describing the role of ZmSTOP1-A in the transcriptional regulation of genes responding to aluminum (Al) toxicity and low-pH stress in maize. Al toxicity and low-pH conditions do not directly induce the transcriptional expression of *ZmSTOP1-A*. Instead, ZmSTOP1-A regulates the expression of genes responsive to Al and low pH. Specifically, ZmSTOP1-A can directly activate the expression of *ZmMATE1* to detoxify Al through citrate secretion. Additionally, ZmSTOP1-A enhances Al tolerance by inhibiting the accumulation of reactive oxygen species (ROS).

## Data Availability

No new data were created or analyzed in this study. Data sharing is not applicable to this article.

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
