# Peer review of "Overexpression of ZmSTOP1-A Enhances Aluminum Tolerance in Arabidopsis by Stimulating Organic Acid Secretion and Reactive Oxygen Species Scavenging"

_ijms, 2023, doi:10.3390/ijms242115669_

Round 1

Reviewer 1 Report

the manuscript ijms-2634576 present s a classical functional study of a gene. In particular, the candidate gene in this case is a transcription factor from Z. mays ZMSTOP1-A, which is supposed to control the plant response to Al toxicity in acidic soils.

The authors have adopted a very careful and elegant experimental plan, exploiting well established techniques. For this reason, the paper does not result particularly novel, albeit correct in principle. On these bases, I suggest the editor to accept the manuscript, even though few aspects should be fixed before publication.

1. Figure 2E is missing

2.the legend of figure 4 is not precise enough. the figure does not result self-explanatory. please add further details. for instance, it is not explained that the panel with plants at pH 5.6 represents the control. it is pretty obvious, but it needs to be mentioned in the legend. In addition, I think that showing the relative root growth in control condition (maybe relative to the wt) could be informative. by looking at the pictures, differences in the primary root growth between the wt, the mutant and the transgenic line are visible. it could be interesting to understand how these differences are impacted by pH and Al-toxicity.

Lines169-170. here there is a bit of chaos about figure 6D. in particular, it seems that figure 6D is mentioned in the text before figure 6C. this has to be fixed, as well as the legend of figure 6.

in the material and methods sections, methods are not explained. the authors use references, but a brief report about the methods used is necessary for further evaluating the techniques applied. 

A general check of the language is suggested

Reviewer 2 Report

Dear Authors,

Reviewer comments ijms-2634576

The manuscript entitled „Overexpression of ZmSTOP1-A enhances aluminum tolerance in Arabidopsis by stimulating organic acid secretion and reactive oxygen species scavenging“ represents a useful study aimed at an investigation of maize ZmSTOP1-A transcription factor conferring enhanced plant tolerance to aluminum and acidic pH stress due to enhanced secretion of organic acids and enhanced ROS scavenging activity. The manuscript includes phylogenetic analysis of ZmSTOP1-like proteins, spatial expression analysis under GUS promoter, subcellular localization of ZmSTOP1, expression analysis under Al and low pH, the effects of ZmSTOP1 expression on the secretion of organic acids and ROS scavenging enzymes (APX, CAT, POD) activities.

I can recommend this manuscript for publication in Internationla journal of molecular sciences.

I have only a few minor comments on the manuscript below:

1/ In Materials and methods, the source of maize inbred line 178 used in the study has to be specified.b The nutrient solution used for plant cultivation and the treatments has also be specified, i.e., the name and composition should be given instead of just references on previous works.

In Results, I would recommend to add a figure providing a model summarising the potential mechanisms of ZmSTOP1 functions in maize response to Al and low pH stress based on the results of the present study.

Formal comments:

Abstract, line 18: Replace the word „completely“ in the statement „Overexpression of ZmSTOP1-A in Arabidopsis Atstop1 mutant partailly restored Al tolerance and enhanced (improved) low pH tolerance…“

Introduction, line 66: Modify the word form „contributed“ to „contributing“ in the statement: „To date, only six genes have been reported contributing to Al tolerance in maize,….“

Line 131: Add the word „possibility“ or „function“ in the statement „To examine this possibility,..“

Figure 5 legend, line 161: Correct the term „Mean values…“ (not „Means values „).

Line 188: Add either the word „result“ or „finding“ in the statement „This result (finding) indicates that SOD, CAT, POD preferntially contribute to scavenging Al-induced ROS,…“

Line 205: Add a comma between the words „cis-elements“ and „respectively.“

Line 206: Correct the spelling of „ABA“ (not „AbA“) for abscisic acid.

Discussion, line 246: Replace the word „But“ with „However“ in the statement „However, in fact, post-transcriptional modifactions…“

Line 268: Modify the word form „predominant“ to „predominantly“ in the statement „I tis notable that STOP1-like genes in most species are predominantly expressed in roots…“

Line 279: Add the word „result“ in the statement „This result suggests ZmSTOP1-A may confer Al tolerance…“

Line 343: Modify the word form „previous“ to „previously“ in the statement: „GUS staining was performed as previously described…“

Line 368: Add the word „previously“ in the statement „…were carried out as previously described…“

Final recommendation: Accept after a minor revision.

Dear Authors,

Reviewer comments ijms-2634576

The manuscript entitled „Overexpression of ZmSTOP1-A enhances aluminum tolerance in Arabidopsis by stimulating organic acid secretion and reactive oxygen species scavenging“ represents a useful study aimed at an investigation of maize ZmSTOP1-A transcription factor conferring enhanced plant tolerance to aluminum and acidic pH stress due to enhanced secretion of organic acids and enhanced ROS scavenging activity. The manuscript includes phylogenetic analysis of ZmSTOP1-like proteins, spatial expression analysis under GUS promoter, subcellular localization of ZmSTOP1, expression analysis under Al and low pH, the effects of ZmSTOP1 expression on the secretion of organic acids and ROS scavenging enzymes (APX, CAT, POD) activities.

I can recommend this manuscript for publication in Internationla journal of molecular sciences.

I have only a few minor comments on the manuscript below:

1/ In Materials and methods, the source of maize inbred line 178 used in the study has to be specified.b The nutrient solution used for plant cultivation and the treatments has also be specified, i.e., the name and composition should be given instead of just references on previous works.

In Results, I would recommend to add a figure providing a model summarising the potential mechanisms of ZmSTOP1 functions in maize response to Al and low pH stress based on the results of the present study.

Formal comments:

Abstract, line 18: Replace the word „completely“ in the statement „Overexpression of ZmSTOP1-A in Arabidopsis Atstop1 mutant partailly restored Al tolerance and enhanced (improved) low pH tolerance…“

Introduction, line 66: Modify the word form „contributed“ to „contributing“ in the statement: „To date, only six genes have been reported contributing to Al tolerance in maize,….“

Line 131: Add the word „possibility“ or „function“ in the statement „To examine this possibility,..“

Figure 5 legend, line 161: Correct the term „Mean values…“ (not „Means values „).

Line 188: Add either the word „result“ or „finding“ in the statement „This result (finding) indicates that SOD, CAT, POD preferntially contribute to scavenging Al-induced ROS,…“

Line 205: Add a comma between the words „cis-elements“ and „respectively.“

Line 206: Correct the spelling of „ABA“ (not „AbA“) for abscisic acid.

Discussion, line 246: Replace the word „But“ with „However“ in the statement „However, in fact, post-transcriptional modifactions…“

Line 268: Modify the word form „predominant“ to „predominantly“ in the statement „I tis notable that STOP1-like genes in most species are predominantly expressed in roots…“

Line 279: Add the word „result“ in the statement „This result suggests ZmSTOP1-A may confer Al tolerance…“

Line 343: Modify the word form „previous“ to „previously“ in the statement: „GUS staining was performed as previously described…“

Line 368: Add the word „previously“ in the statement „…were carried out as previously described…“

Final recommendation: Accept after a minor revision.
